cognition, behaviour

responsibility, joining groups, loss, regret, negative outcomes

**Author for correspondence:**
Marwa El Zein
e-mail: marwaelzein@gmail.com, m.zein@ucl.ac.uk

# Joining a group diverts regret and responsibility away from the individual

Marwa El Zein[1,2] and Bahador Bahrami[2,3,4]

[1]Institute of Cognitive Neuroscience, University College London, Alexandra House, 17–19 Queen Square, London WC1N 3AZ, UK
[2]Center for Adaptive Rationality, Max Planck Institute for Human Development, Lentzeallee 94, 14195 Berlin, Germany
[3]Faculty of Psychology and Educational Sciences, Ludwig Maximilian University, Leopoldstrasse 13, 80802 Munich, Germany
[4]Department of Psychology, Royal Holloway, University of London, Egham, Surrey TW20 0EX, UK

ME, 0000-0002-6052-6210

It has recently been proposed that a key motivation for joining groups is the protection from the negative consequences of undesirable outcomes. To test this claim, we investigated how experienced outcomes triggering loss and regret impacted people's tendency to decide alone or join a group, and how decisions differed when voluntarily made alone versus in group. Replicated across two experiments, participants ($n = 125$ and $n = 496$) selected whether to play alone or contribute their vote to a group decision. Next, they chose between two lotteries with different probabilities of winning and losing. The higher the negative outcome, the more participants switched from deciding alone to with others. When joining a group to choose the lottery, choices were less driven by outcome and regret anticipation. Moreover, negative outcomes experienced alone, not part of a group vote, led to worse subsequent choices than positive outcomes. These results suggest that the protective shield of the collective reduces the influence of negative emotions that may help individuals re-evaluate past choices.

## 1. Introduction

When things go wrong, people often look for someone to blame. From war tribunals and post-mortem inquiry committees investigating global disasters to brooding over coffee the day after a failed dinner date, people strive to find out who or what to blame. What is often forgotten is the uncertainty under which decisions had been made. In retrospect, the negative outcomes often seem to have been the deterministic consequences of the decisions. Decision-makers therefore seek to protect themselves from possible future blame for difficult uncertain decisions. A doctor who has documented her colleagues' confirmation of a diagnosis is better able to defend herself against both regret and charges of malpractice. Intelligence analysts who held that the weapons of mass destruction existed in Iraq in 2003 still cite the consensus among various agencies as their strongest excuse. Indeed, circumstantial evidence from numerous studies in social and cognitive psychology suggests that when facing decisions with uncertain outcomes, being in a collective can help protect individual members from negative consequences [1]. Here we directly test this hypothesis in an empirical investigation that demonstrates how experienced outcomes change the propensity to join groups, which may protect from anticipated and experienced consequences of negative outcomes.

Previous studies in decision-making using behavioural [2–4] and psychophysiological measures [2] have shown that negative outcomes trigger strong negative emotions, such as loss after negative factual outcomes (i.e. when a selected option results in a negative outcome) and regret after negative counterfactual outcomes (i.e. when an unselected option results in a better outcome than the

selected one). Moreover, individuals choose not only options that optimize their choice by maximizing the expected value, but also those that minimize future regret [2–7]. The experience of negative emotions associated with costly decisions is related to the responsibility for those decisions: emotional responses to losses are stronger if people are responsible for their actions [3]. Furthermore, feeling responsible has been suggested as a prerequisite to feeling regret about an outcome [8].

We recently hypothesized that, by distributing the responsibility for decision outcomes among several group members, making decisions as part of a group could help regulate negative emotions associated with negative factual and counterfactual outcomes [1]. Supporting this claim, choosing a lottery as a part of a group reduces the feeling of responsibility and regret over negative outcomes [9]. Moreover, anticipating regret leads people to delegate difficult decisions to others [10] and use institutions strategically in order to share responsibility with others when trading lottery tickets [11]. It remains unknown; however, how negative emotions influence the choice to make costly decisions alone or with others, and how these voluntarily chosen group decisions may differ from individual ones.

Here we investigate how negative outcomes influence participants' preference to make decisions alone or as part of a group. Using a task in which decisions were costly and could elicit highly negative emotions via factual (loss) and counterfactual (regret) outcomes [3], we asked participants to choose, on a trial-by-trial basis, whether to decide alone or contribute their opinion to a majority vote-count in a group of five. Such majority votes offer an empirically controlled way of sharing responsibility with other individuals for a choice's outcome. Using computational behavioural analysis, we assessed whether the previously established influence of expected value and anticipated regret on individual lottery decisions changes when the participant acted as a member of a group. Importantly, our design offered a powerful quantitative method to test how the valence and magnitude of experienced outcomes, combining loss and regret, influenced participants' choices to decide alone or in a group.

Based on our hypothesis that being a member of a group can protect individuals from the negative consequences of decisions [1], we predicted that (i) for a person making a decision alone, the more negative their experience from an outcome, the higher the likelihood of them joining a group for the next decision; (ii) when individuals vote as part of a group, their choice would be less strongly driven by anticipated outcomes than when deciding privately; and (iii) experienced negative outcomes would disrupt subsequent value-based choices less strongly if experienced in a group as compared to individually.

## 2. Material and methods

### (a) Participants

In experiment 1, data were collected using Amazon's Mechanical Turk and included a total of 125 participants (mean age 31.72 ± 7.31, 88 males). In experiment 2, data were collected using prolific (www.prolific.co) and included a total of 502 participants initially. Six participants were excluded from analyses because they responded too fast (mean reaction time less than 200 ms), and/or missed more than half of the trials, and/or consistently pressed the same left or right button for lottery choice (more than 90% of

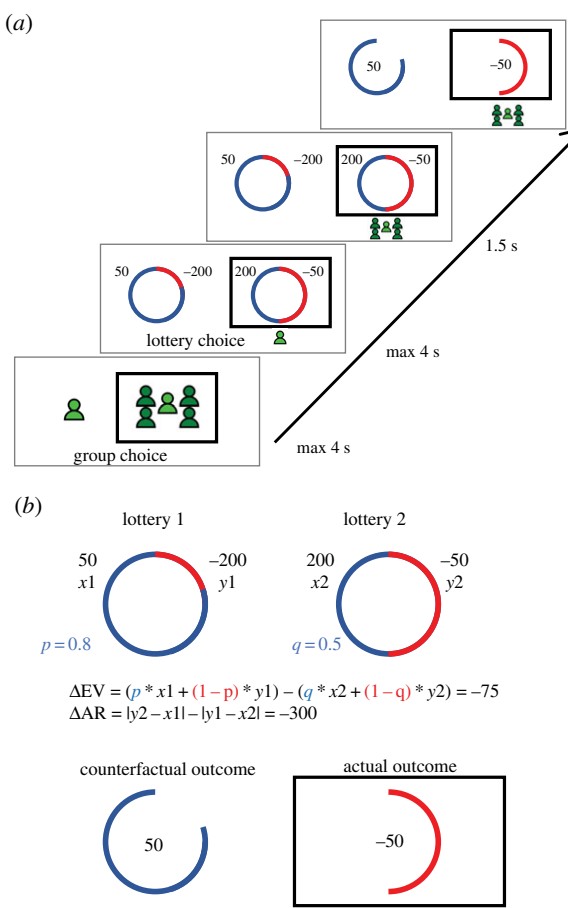

**Figure 1.** Experimental design. (*a*) Timeline of the experiment. First, participants chose whether to play alone or in a group. Second, they chose between two lotteries. Both factual and counterfactual outcomes were shown. (*b*) Description of parameters of interest. The size of coloured sectors of the circle represents the different probabilities of winning or losing. $p$ and $q$ are the probabilities of getting the best outcomes, $x1$ (lottery 1) and $x2$ (lottery 2), respectively. $y1$ and $y2$ represent the worst outcomes for lotteries 1 and 2, respectively. Predictors of lottery choice: $\Delta$EV corresponds to the difference in expected value between lottery 1 and lottery 2. $\Delta$AR represents the difference in anticipated regret between the two lotteries; the higher this difference, the more choosing lottery 1 over lottery 2 allows for minimizing future regret. Predictor of the group choice: experienced outcome combines both actual and counterfactual outcomes to quantify the amount of experienced positive or negative outcomes at each round. (Online version in colour.)

the time), leaving 496 participants (mean age 31.49 ± 11.09, 215 females). The experiment lasted around 10 min and each participant was paid $2 + $1 bonus in experiment 1 and £1.5 + up to £1 bonus in experiment 2.

### (b) Experimental design and task

#### (i) Timeline

Participants completed a value-based decision-making task adapted from Coricelli *et al.* [3]. At each trial, they had to choose between two lotteries where the size of coloured sectors of a circle represented the probabilities of winning or losing (blue sectors depicted the chance of getting the best outcome and red sectors depicted the chance of getting the worst outcome; figure 1*a*). Choices were registered by pressing either the F key (for the lottery on the left, lottery 1) or the K key (for the lottery on the right, lottery 2). In each round, before choosing between the lotteries, participants had to decide whether to play the round alone or in a group with four other players (selected by pressing the F or K

key, this mapping was counterbalanced across participants in experiment 2). Choice time for both decisions was limited to 4 s each. Participants were told that if they did not answer within that time they would receive the worst possible outcome in that round.

Once a participant selected a lottery, a black square framed the selection and an image of an individual appeared underneath. Participants in a group saw a black square around the majority option with an image of five individuals underneath. The outcomes of both lotteries were shown for 3 s (figure 1a). After eight practice trials, each participant completed four blocks of 12 trials each.

### (ii) Majority vote procedure

When deciding to play in a group with four other players, participants were told that the chosen lottery depended on a majority rule: the lottery chosen by three or more players would be selected.

In experiment 1, the confederate votes were drawn from the behaviour of 11 participants who, in an earlier pilot, had undertaken the same task under alone condition only. This pilot was conducted on Amazon's Mechanical Turk (mean age $41.5 \pm 14$, five males). For each pair of lotteries, the lottery that was chosen by more than five participants was selected as the group choice.

In experiment 2, to avoid deception and to allow for more variability in the group decisions, the majority vote was based on the participant's choice and the choices of randomly drawn samples from experiment 1 data for the same lottery pair. For each pair of lotteries, the lottery that was chosen by more than three participants (including the participant's choice) was selected for the round.

This design allowed us to maintain control over 'other's behaviour' as an independent variable. It is important to underscore here that the aggregation of the votes of such virtual groups does not constitute an emergent collective. The design we chose here is appropriate to test our hypothesis about how the context of deciding alone versus with others affects individual choices but, and importantly, our hypothesis does not involve examining the emergent collective behaviour of a group of interacting individuals.

### (iii) Payment and incentive

At the beginning of the experiment, participants were told they would receive \$2 (experiment 1) or £2 (experiment 2) for their participation and that one round would be randomly selected so that they could earn up to an additional \$1 (experiment 1) or £1 (experiment 2), depending on the outcome of that round. In experiment 1, we accorded the \$1 bonus to all participants. In experiment 2, one round was indeed randomly selected and participants received a bonus of £1 if they received the highest outcome (200), £0.25 if they received the second highest outcome (50), and no bonus if they hadn't answer on that round or received a negative outcome.

### (iv) Lottery structure

The lotteries presented were identical to the 48 trials previously used by Coricelli et al. [3]. The same pairings of combinations of outcomes (with values of 200, 50, −50 or −200) with three possible outcome probabilities (0.2, 0.5 and 0.8) were used (see the electronic supplementary material, table 1). Actual probabilities in each round corresponded to the displayed probabilities. Pairs of lotteries were shown in a randomized order for each participant.

## (c) Behavioural modelling of choice and statistical analyses

There were two choices of interest: the lottery choice (lottery 1 or lottery 2, based on probabilities and outcomes) and the group choice (play alone or in a group, in each round before seeing the two lottery options; figure 1). To compute what significantly influenced lottery choice and group choice, while accounting for individual variability, we used mixed-effects logistic regression models implemented with maximum penalized likelihood (MPL) via the glmer function in the R package lme4 [12]. In the performed mixed models, subjects were modelled as a random intercept and the propensity to play alone as random slope. The parametric regressors were scaled before being entered in the models. The regressors used to explain both the types of choices are detailed below.

### (i) Group choice at time $t$

Four predictors at time $t-1$ (previous round): valence, magnitude, previous condition, and group status, were computed as follows.

First, the experienced outcome at $t-1$ was computed as a combination of factual and counterfactual: obtained outcome–unobtained outcome from unchosen lottery. Negative values of experienced outcome correspond to increasing loss and regret, while positive values correspond to increasing sense of success and relief. We then separated the valence of the outcome from the magnitude of experienced outcome.

— For the valence predictor, these values were sign transformed to obtain negative (−1) versus positive outcomes (+1).
— For the magnitude predictor, we transformed experienced outcome values into magnitude values only reflecting the change in magnitude between the outcomes. Outcome values (−400, −250, −150, −100, 100, 150, 250, 400) were transformed for negative values into (0, 150 250, 300) and for positive values into (0, 50, 150, 300).

Negative values:

   (i) 0 + (150 difference between −400 and −250)
   (ii) 50 + (100 = difference between −250 and −150)
   (iii) 250 + (50 = difference between −150 and −100)

Positive values:

   (i) 0 + (50 difference between 100 and 150)
   (ii) 50 + (100 = difference between 150 and 250)
   (iii) 150 + (150 = difference between 250 and 400)

Owing to the lottery structures, the magnitude of possible outcomes was slightly different for positive and negative outcomes.

— Previous condition: whether the round is played alone or in a group (and thus whether outcome is experienced alone or as part of a group).
— Group status: when the condition is in a group, whether the participant is in the minority or the majority (for the lottery choice based on a majority rule).

### (ii) Lottery choice at time $t$

Two predictors of lottery choice (choosing lottery 1) were computed as in Coricelli et al. [3]: expected value and anticipated regret.

— Difference in expected value between lottery 1 and lottery 2:

$$\Delta EV = EV1 - EV2,$$

where EV is:

$EV1 = px1 + (1 - p)y1$   (lottery 1)
$EV2 = qx2 + (1 - q)y2$   (lottery 2).

$p$ and $q$ represent the probability of receiving the highest outcome $x1$ or $x2$ (in blue; figure 1) for lottery 1 and lottery 2, respectively. $y1$ and $y2$ denote the lowest outcome from lottery 1 and lottery 2, respectively (in red). The higher the difference

in expected of value between lottery 1 and lottery 2, the more the participant should choose lottery 1.

— Difference in anticipated regret between lottery 1 and lottery 2:

$$\Delta AR = AR2 - AR1,$$

where AR is:

$$AR1 = |y1 - x2| \quad AR2 = |y2 - x1|.$$

Anticipated regret is represented by the difference between the lowest and the highest outcome across lotteries. If the $\Delta AR$ is positive, receiving the worst outcome from lottery 2 while lottery 1 results in its best outcome will elicit higher regret than would receiving the worst outcome from lottery 1 while lottery 2 results in its best outcome. Therefore, the higher the value of $\Delta AR$, the more anticipated regret would be avoided by choosing lottery 1. Note that while the expected value depends on the probabilities related to the lotteries, anticipated regret depends solely on outcomes values.

The interaction of these predictors with the current condition (playing alone/in group at time $t$) was also entered in the model predicting lottery choice. For the analyses on how experienced outcome at $t-1$ influenced lottery choices at time $t$, previous condition, valence and group status ($t-1$) were entered as predictors of lottery choice in addition to $\Delta EV$ and $\Delta AR$ ($t$) of that lottery pair.

$\Delta EV$ and $\Delta AR$ are correlated choice predictors ($r = 0.75$, $p < 0.001$); however, (i) their variance inflation factor is equal to 2.33 suggesting that it is acceptable to put them in the same general linear model. In the models, only the variance not accounted by the other predictor is reflected in the results. (ii) The parametric regressors were QR Gram Schmidt orthogonalized (using QR() and QR.Q functions in R) to enter in our generalized linear model the residuals of the regressors after removing the common variance to both predictors. This yielded to exactly similar results (see the electronic supplementary material, table 7) confirming that it is correct to include them both as predictors of the lottery choice, and showing that this orthogonalization procedure was performed automatically via the glmer R function.

Confidence intervals of parameter estimates of the mixed models are reported and were calculated using the function *confint* in R. The metafor package in R [13] was used to perform meta-analytic analyses combining both experiments via the function *rma* (to assess what are the consistent results across both experiments).

For clarity of presentation of the results, descriptive results are shown on figures, with standard errors of proportions (SEP) calculated as

$$SEP = \sqrt{(p \times (1 - p)) \div n},$$

where $p$ is the proportion choosing lottery 1 (versus lottery 2) or playing in a group (versus alone) and $n$ is the number of observations.

# 3. Results

Following our three predictions, we first consider what influenced the decision to join a group or decide alone. The second and third sections examine the determinants of the lottery choice.

## (a) When to join a group decision

In each round, before seeing the lotteries, participants first chose whether they wanted to play an upcoming lottery choice alone or in a group. On every trial, this decision to join the group or not was made before the lotteries were seen. This ensured that we could assess the direct influence of previous experienced outcome on this choice independently from the upcoming lottery structure. Participants showed highly diverse patterns of behaviour in their tendency to play alone or with a group (figure 2a). It is important to note that about half of participants consistently chose to play alone or in group (more than 90% of the times—experiment 1, 60 out of 125; experiment 2, 269 out of 496) and therefore were not affected by experienced outcomes.

Asking participants to choose whether to play alone or in a group at the start of each trial allowed us to quantify the impact of the magnitude and the valence of experienced outcomes on the probability of joining the group in the next trial (figure 1b). This measure combines factual and counterfactual outcomes, with no aim of dissociating between the influence of one or the other, but rather focusing on a global valence measure (i.e. the magnitude of negative and positive experience) with which we could test our predictions. We reasoned that the more negative experienced outcomes, the higher increased loss and regret by comparing the actual outcome to a more positive counterfactual outcome and vice versa, the more positive the outcome, the higher the positive experiences of success and relief. We then define two regressors to dissociate the *valence* (positive versus negative experience) from the *magnitude* (amount of experienced loss/regret and success/relief) of the outcome (see methods for details).

We first tested our prediction that a negative outcome experienced alone will increase the likelihood of joining a group for the next decision [1]. We therefore considered another regressor indicating the participants' previous condition—that is, whether they were playing alone or in a group. We examined the effect of previous condition and valence, and their interaction on the choice to join a group on the next trial using mixed-effects logistic regression models. A main effect of previous condition (see full statistical details in the electronic supplementary material, tables 2, 3 and 4, all $z > 8.96$, all $p < 0.001$), reflected the fact that some participants chose to play predominantly alone or in a group (as shown in figure 2a), but also a sort of inertia effect whereby people stick to their previous decision. Indeed, even when excluding participants who invariably chose to stay alone or in group—the main effect of previous condition remained significant (see the electronic supplementary material, table 5, meta-analytically $z = -3.9$, $p < 0.001$).

### (i) Valence effects

An interaction between previous condition and valence (all $z > 4.79$, all $p < 0.001$, see the electronic supplementary material, tables 2, 3 and 4, but also electronic supplementary material, table 8 for results after excluding participants who consistently chose to play alone or in group) revealed a win-stay lose-change effect: participants were more likely to switch their choice after a negative versus a positive outcome experienced alone (all $z > 3.11$, all $p < 0.04$, also see individual difference information on the propensity to switch from alone to group based on valence in the electronic supplementary material, figure 1), or in a group (all $z > 3.57$, all $p < 0.002$) (figure 2b). This confirms our first prediction that people are more likely to join groups after individually experiencing negative outcomes. Next, we asked if the likelihood of joining groups changed linearly with the magnitude of experienced outcomes.

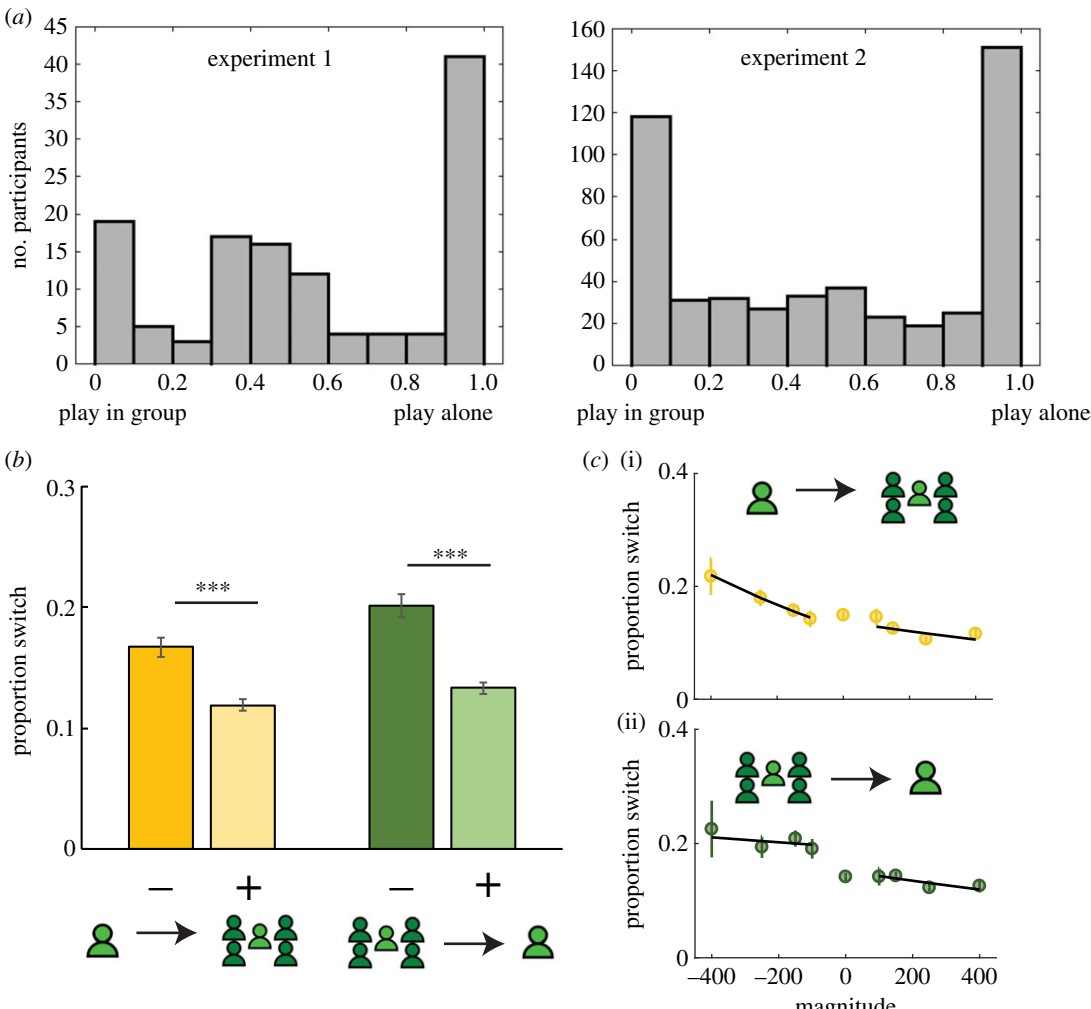

**Figure 2.** Group choice. (*a*) Histogram showing participants' overall tendency to play alone or in a group in experiment 1 (left panel) and experiment 2 (right panel). (*b*) In experiment 2, proportion of rounds where participants switched their choice to play alone or in group as a function of valence in the previous round: − negative outcomes, + positive outcomes, and of previous condition: outcomes experienced alone (yellow), outcomes experienced in group (green). (*c*) In experiment 2, proportion of rounds where participants switched their group choice as a function of outcome magnitude. (i) Outcome experienced alone. (ii) Outcome experienced in a group. Error bars represent standard errors of proportions. Statistics on the figure correspond to the reported outputs of the mixed models, ***$p <$ 0.001. (Online version in colour.)

### (ii) Sensitivity to the magnitude of outcomes

An interaction was observed between previous condition and magnitude (meta-analytically across both experiments ($z > 1.96$, $p < 0.05$,). The choice to join the group was sensitive to the magnitude of outcomes experienced alone (meta-analytically $z = 2.36$, $p = 0.01$ − experiment 1 $p = 0.17$, experiment 2 $p = 0.04$). In other words, the less positive/more negative the outcome, the more likely were the participants to join a group on the next round. By contrast, when in group, the magnitude of outcomes did not change the propensity to join the group in the next trial (all $p > 0.26$) (figure 2c). Thus, when participants played alone, they were sensitive to the magnitude of outcomes beyond the win-stay lose-change effect, to guide their choice to join the group.

We then turned to our next question: once in the group, what makes people stay or leave the group above the win-stay lose-change effect?

### (iii) Groups status and outcomes

Being in a group is a compromise between retaining full autonomy for a choice and giving up autonomy completely [1]. In our study, this compromise was achieved when participants'

vote corresponded to the majority vote, when they shared responsibility with group members for decision outcomes. We predicted that when supported by majority, people would stay in the group no matter the outcome, as they would be protected from loss and regret by shared responsibility. However, when in the minority and not responsible for the outcome, we predicted that participants would be more likely to leave the group following negative outcomes.

Being in majority or minority influenced the decision to play alone or in group, as well as experienced outcome effects on this decision. Indeed, we examined the effect at $t − 1$ of group status (majority/ minority), valence and magnitude, and their interaction on the choice to join a group at $t$ when considering only previous group trials. Participants were more likely to choose to play in group again if they were part of the majority i.e. agree with the group decision, as compared to minority (all $z > 2.5$, all $p < 0.01$) (electronic supplementary material, figure 2a). Agreeing with the group may also protect from the influence of outcome valence on the decision to play alone or in group: an interaction between group status and valence appeared in experiment 1 (meta-analytically $z = 1.67$, $p = 0.09$, experiment 1 $p = 0.0413$, experiment 2 $p = 0.25$), suggesting that participants may be more sensitive to valence

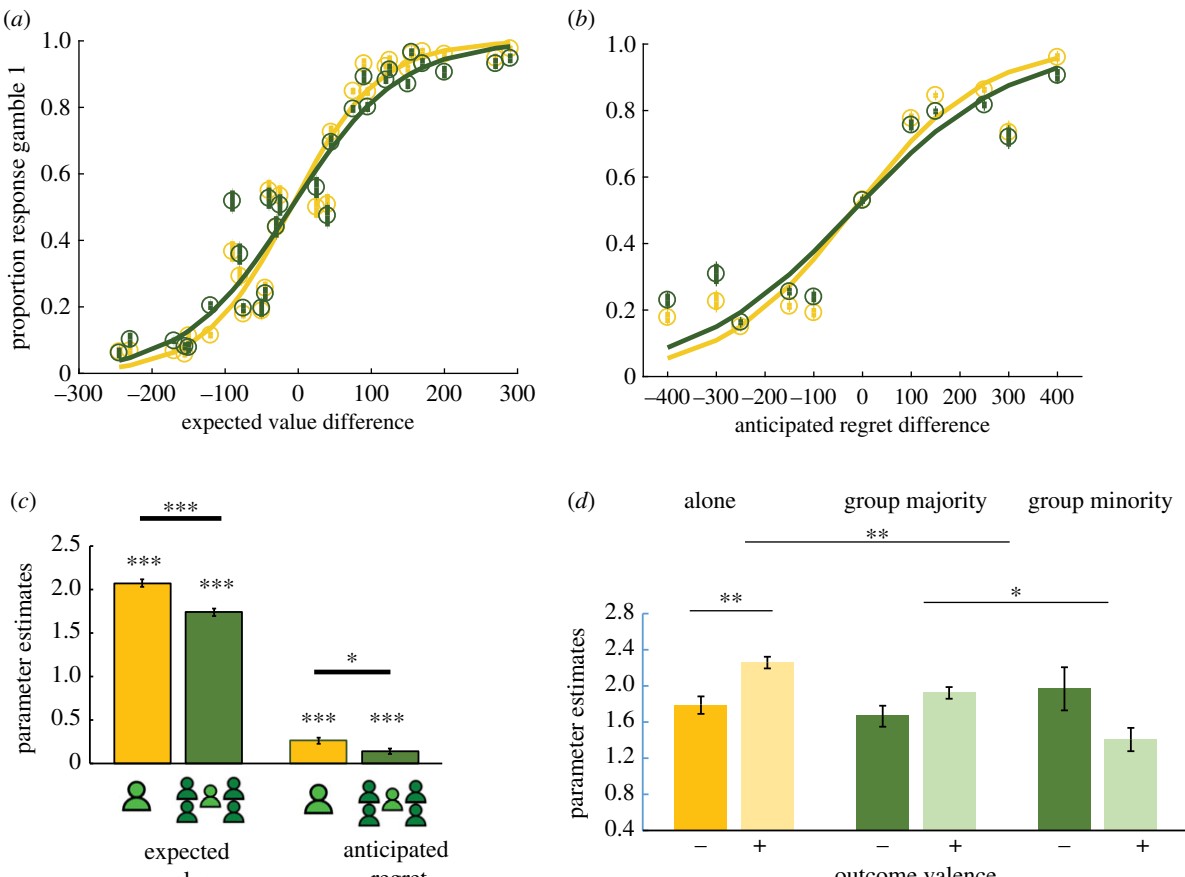

**Figure 3.** Lottery choice. (*a*) Proportion of choosing lottery 1 versus lottery 2 based on the difference in expected value ΔEV in experiment 2. (*b*) Proportion of choosing lottery 1 versus lottery 2 based on the difference in anticipated regret ΔAR in experiment 2. (*a,b*) Yellow represents the alone condition while green represents the group condition. Error bars represent standard errors of proportions. (*c*) Parameter estimates of expected value (ΔEV) and anticipated regret (ΔAR) in mixed models separately for trials where participants played alone (yellow) or in a group (green). (*d*) Parameter estimates of expected value (ΔEV) in mixed models separately following trials where participants experienced negative (−) and positive (+) outcomes on the previous round. From left to right: outcomes experienced alone (yellow), outcomes experienced in group majority (green), outcomes experienced in group minority (green) (*c,d*). Error bars represent the standard error of the estimated parameters in the mixed model. *$p < 0.05$, **$p < 0.01$, ***$p < 0.001$. (Online version in colour.)

after minority as compared to majority (electronic supplementary material, figure 2b).

Thus far we saw that negative versus positive outcomes drive people to switch from playing alone to playing in a group. Moreover, this change was sensitive to the magnitude of the outcomes. We next asked if when in a group, the individual would be less likely to worry about regret.

## (b) Influence of anticipated outcomes on lottery choices made alone or in group

Previous studies have shown that when deciding individually, anticipated regret could impair risky decision making in individuals. We asked whether this detrimental impact of anticipated regret on value-based choice would be reduced when participants choose as part of a collective (versus individually). We considered two predictors of the lottery choice: expected value (ΔEV) and anticipated regret (ΔAR; figure 1b; see methods for details), and their interaction with whether participants chose to play alone or in group at $t$ (current condition, lottery alone/lottery in group). Mixed-effects logistic regression models showed an additive influence of expected value (all $z > 25.03$, all $p < 0.001$, see the electronic supplementary material, tables 2, 3, 4) and anticipated regret (all $z > 2.26$, all $p < 0.05$). This result established that participants minimized future regret when choosing between the two lotteries (figure 3a,b).

Importantly, current condition (alone/in group) interacted with both expected value (all $z > 3.24$, all $p < 0.002$) and anticipated regret (meta-analytically $z = −2.29$, $p = 0.02$, experiment 1 $p > 0.5$, experiment 2 $p = 0.01$), showing that ΔEV and ΔAR parameter estimates were both reduced in the group as compared to alone condition (figure 3a,b,c).

These results support our hypothesis that contributions to group decisions are less driven by expected value and regret than individual decisions. Joining groups can therefore reduce the consideration of *anticipated* outcomes associated with emotions of loss and regret. In the next section, we test the hypothesis that negative experienced outcomes leave less of an adverse emotional impact on people's decision making if they were experienced as a group versus alone.

## (c) Influence of experienced outcomes on lottery choices

To test our final prediction that experienced negative outcomes disrupt subsequent lottery choices less strongly if experienced after a group versus individual decision, we asked how experienced outcomes impact the next lottery choice. We entered (i) valence at $t−1$ and (ii) previous condition (at $t−1$) (alone or in group), as predictors of the lottery choice at trial $t$ in our mixed model, in addition to ΔEV($t$) and ΔAR($t$). In other words, we examined whether experienced outcomes from

one round affect the influence of expected value and anticipated regret on the lottery choice in the next round. An interaction between previous condition, valence and $\Delta$EV was observed (all $z > 2.14$, all $p < 0.05$). This interaction (also see additional interpretation of the triple interaction in the electronic supplementary material) showed that valence influenced $\Delta$EV, only when the outcome was experienced alone (interaction valence*$\Delta$EV all $z > 2.835$, all $p < 0.004$), not in group (all $z < 1.350$, all $p > 0.177$). Negative versus positive outcomes reduced the $\Delta$EV parameter for the next lottery choice, only when experienced alone (figure 3$d$), but critically not in group (see next paragraph on how this is different based on group status). There was no significant change in $\Delta$AR parameter based on valence and previous condition (all $p > 0.1$). These results show that negative outcomes experienced alone disrupted the subsequent lottery choice by reducing reliance on expected value to guide choices. However, this was not the case when negative outcomes were experienced as a group.

### (i) Responsibility modulates the adverse impact of negative outcome

Is the influence of negative outcomes on future lottery choices modulated by responsibility? As we argued earlier, our experimental paradigm permits investigating the role of responsibility by contrasting behaviour when participants were in the group majority versus minority. We focused on trials where participants chose to vote in group and entered as predictors of the lottery choice the following variables: group status $(t - 1)$ and valence$(t - 1)$ and their interaction with expected value $\Delta$EV$(t)$ and anticipated regret $\Delta$AR$(t)$. An interaction between group status, valence and $\Delta$EV was observed (meta-analytically $z = -2.79$, $p = 0.005$, experiment 1 $p = 0.05$, experiment 2 $p = 0.02$), showing that only when people were in group majority was their subsequent choice unaffected by negative outcome (interaction $\Delta$EV*valence all $p > 0.11$). However, after experiencing the negative outcome as minority (interaction $\Delta$EV*valence after minority meta-analytically $z = -1.89$, $p = 0.05$, experiment 1 $p = 0.02$, experiment 2 $p = 0.10$), $\Delta$EV parameter was increased, not reduced, after negative versus positive outcomes (figure 3$d$).

## 4. Discussion

Our novel experimental design in which participants were given the choice to play alone or in a group on a trial-by-trial basis provides, for the first time to our knowledge, a cognitive basis for the motivation to join groups. We show that the higher their negative experience, the more likely participants were to join a group, consistent with decreasing the burden of individual responsibility and blame [1]. Accordingly, joining the group reduced the impact of anticipated regret on choices. Moreover, contrary to individually experienced negative outcomes, experiencing negative outcomes as a member of group majority did not impair subsequent lottery choice as compared to positive outcomes. These results suggest that voting in a group rendered people less vulnerable and responsive to anticipated and experienced outcomes.

We note that our design was not meant to and did not permit accounting for the collective behaviour of interacting individuals. Our objective here was to take the individual's perspective to address what motivates people to join a group. Therefore, we chose a majority vote procedure using previously gathered responses from participants in a separate experiment thereby maintaining control over other 'group members' behaviour and allowing us to address majority and minority impact on the choice to play alone or join a group. Future research is necessary to examine whether our results replicate under the less controlled design of an entire group of autonomous interacting individuals.

It has been suggested that responsibility rests on perceiving oneself as the agent of an action and believing that one could have done otherwise [14]. Both conditions were met here when individuals played alone: they were the only agent and had access to counterfactual outcomes and therefore were highly responsible for the outcome. Joining a group to diminish responsibility over decision outcomes could help alleviate negative emotions such as loss and regret [1]. Accordingly, participants were more likely to join groups after negative versus positive outcomes and were sensitive to the amount of experienced outcomes. These results are consistent with previous studies on delegation showing that people prefer to give up autonomy when faced with difficult choices, choices they might regret, or decisions with high risk of error [5,10,15–18].

In parallel to the influence of valence, participants tended to repeat their choice to stay alone, revealing an inertia/status-quo effect [19]. This was also the case when people were playing in group. Moreover, the tendency to repeat one's choice was observed in participants who regularly switched their choice from playing alone to playing in group. These results suggest that (i) there are baseline, trait-like preferences to play alone or in group that are independent from experienced outcomes and (ii) even when people vary their choice, a tendency to repeat the previous choice may show that participants exploit the same option before switching. Interestingly, the emotion of regret may induce a status-quo bias [9]—suggesting that status quo in the decision to play alone or in group may be a strategy to reduce regret in the context of the current study.

Our main prediction in this paper was that being in a group will help protect against the negative consequences of decisions [1]. Confirming this prediction, we show that joining a group after negative outcomes has a protective role: participants were less influenced by future regret and outcomes when picking a lottery in a group as compared to when acting alone. From the shared responsibility hypothesis perspective [1], this might suggest that participants feel less responsible for their choice when playing in a group. Relatedly, previous studies have shown that subjective ratings of responsibility and regret for probabilistic outcomes are reduced when people play lotteries collectively rather than individually [9,20]. Regret has been considered as a form of automatic self-punishment [21], based on a comparison of one's own actions and what is accepted by the group. Following this conception of regret, collective decisions would be favoured as they would naturally reduce self-punishment. Additionally, given that expected value influence also decreased in groups, participants may have chosen to play in the group to exert less effort—a phenomenon known as social loafing [22].

When already part of a group, agreement with the group majority predicted the decision to stay in group on the next round. Even in our experiment's minimal group set-up where the other group members were totally unknown to participants, they showed a bias towards staying within a group that supported their decisions. Possibly, simply agreeing with the group creates an in-group favouritism such as in minimal group paradigms [23], which favoured playing with the group

again. Indeed, neuroscientific evidence [24,25] for motivational value of like-minded others' agreement and social support for our preferences points to the importance of group endorsement as a strong motivational driver of choice behaviour.

A difference in the influence of experiencing outcomes was observed that depended on the participant's majority versus minority status: while experiencing negative (versus positive) outcomes in the minority led to an increased optimization of choice in the next lottery choice (increased influence of expected value), no such difference was observed after experiencing outcomes in majority. Being in the group majority also reduced the sensitivity to the valence of experienced outcomes for the next group choice (significant only in experiment 1, $p = 0.09$ meta-analytically). Being in a group could promote a sense of joint agency. Joint agency has been described as 'we-mode' and consists of a shift from self-agency to we-agency in collective actions [26–28]. Individual decision-makers might be operating in different modes when they are in the majority and thus perceiving their group as an entity 'we-mode' [27] versus when they are in the minority and see their group as a collection of isolated individuals. Our findings could be interpreted as evidence that when people are in this 'we-mode,' calculations of loss and regret may become less relevant.

Altogether, our findings suggest that people may be less sensitive to both anticipated and experienced outcomes when they vote as a group. Interference arising from the social rewards of finding oneself in a group may prevent people from learning from feedback or even reduce their motivation to seek feedback. In line with this view, people do less fact-checking when they encounter suspicious information in social digital environments while in the presence of others than if they are alone [29]. Furthermore, people are less likely to integrate new information that contradicts the beliefs of their group [30]. Together with our results, these findings suggest that being in a group might interfere with the integration of new information—for example, from feedback. This idea has far-reaching implications for understanding the impact of social context on political attitudes. Common

sense requires that individuals reflect on previous actions. Our findings suggest that this may be less likely to occur when negative outcomes have resulted from a vote within a group. This helps explain why partisan allegiances are resilient [30] to outright the evidence of incompetence in political leadership, incriminating and scandalous information about politicians, and even catastrophic outcomes of failed policies that were supported by the majority.

In conclusion, experience related to factual and counterfactual outcomes influenced people's propensity to make decisions alone or in a group: experiencing high negative outcomes pushed participants towards joining decisions in a group that benefited from reduced anticipated regret. Negative versus positive outcomes experienced alone disrupted future choices, while those experienced as a group majority protected from such influence. These findings offer important insight into questions such as whether belonging to a political party renders its members less careful and responsive to the party's anticipated or experienced failures and successes.

Ethics. All participants provided consent according to regulations approved by the UCL Research Ethics Committee (project ID number 5375/001).

Data accessibility. Data is available from the Dryad Digital Repository: https://doi.org/10.5061/dryad.gqnk98shr [31].

Authors' contributions. M.E.Z and B.B. both designed the study. Coding the experiment, data collection and data analyses were performed by M.E.Z. Suggestions of data analyses and interpretation of results were done by both authors. M.E.Z drafted the manuscript. B.B. provided important revisions. Both authors approved the final version of the manuscript for submission.

Competing interests. We declare we have no competing interests

Funding. M.E.Z. is supported by the Wellcome Trust (grant no. 204702). B.B. is supported by the European Research Council (ERC) under the European Union's Horizon 2020 research and innovation programme (grant no. 309865; acronym: NEUROCODEC; grant no. 819040; acronym: rid-O). B.B. was also supported by the Humboldt Foundation (GBR 1144057 HFST-E) and by the NOMIS foundation.

Acknowledgements. We would like to thank Deborah Ain for editing the manuscript.

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
