## [Reviewer comments · Proceedings of the Royal Society B: Biological Sciences]

Review History

RSPB-2019-2251.R0 (Original submission)

Review form: Reviewer 1

Recommendation

Accept with minor revision (please list in comments)

Scientific importance: Is the manuscript an original and important contribution to its field?

Good

General interest: Is the paper of sufficient general interest?

Good

Quality of the paper: Is the overall quality of the paper suitable?

Good

Is the length of the paper justified?

Yes

Should the paper be seen by a specialist statistical reviewer?

No

Do you have any concerns about statistical analyses in this paper? If so, please specify them explicitly in your report.

No

It is a condition of publication that authors make their supporting data, code and materials available - either as supplementary material or hosted in an external repository. Please rate, if applicable, the supporting data on the following criteria.

Is it accessible?

Yes

Is it clear?

Yes

Is it adequate?

Yes

Do you have any ethical concerns with this paper?

No

Comments to the Author

The authors report the results of an interesting experiment on the choice to either make choices in isolation or be part of a collective decision. Participants were asked to choose between two risky gambles. Experienced and anticipated regret played a role in the decision to be part of a collective choice.

This is an interesting and original experimental work.

Comments:

My main concern is about the fact that the behavior of others in the experiment was not a collective behavior, thus they didn't know that they were part of a group. This would have potentially affected their behavior. Thus, the participants are observing the sum of individual and not collective behavior.

"We note here that our aim is not to disentangle regret and loss from one another; instead we acknowledge that they are strongly related negative emotions, and focus on examining their hypothesized impact on the choice to join a collective." This sentence is not clear, Please rewrite it in a more comprehensive way

In experiment 1 participants experienced choices made by others in a one-player condition. This might have affected the data. A comparison of the results from Exp1 and Exp 2 is missing.

Experiment 2, not clear "when deciding to play in a group or alone and in when they decided to play in group what was their choice? Motivation for this is not clear. Do they choose before or after observing the choice of the others? Is their choice part of the collective choice?"

How different is the behavior of the group compared with the one of each participant and how this correlates with the choice to join the collective. For each choice parameter, i.e. ev, regret and risk, you should estimate individual differences with the group.

Experiment 2, I would suggest to show the choice data of the reference group, i.e. estimate how many risky vs. safe choices they made independently from the individual choice (i.e. with already 3) and the same analysis for anticipated regret choices.

Do participants first see the lotteries and then decide if join or not the group? This would have

been a nice extension of the study and would have linked this study with the literature about confidence.

Group choice. Not clear if COV and COM are lagged variables, i.e. t-1. Not clear how you computed the COM. So choice at time t as a function of lagged variables. It's confusing because you call the "most recent outcome" the current outcome, and then use it as a lagged variable. This section is very confusing. You can still disentangle the effect of the outcome and the effect of the counterfactual outcome on subsequent choice. E.g. the participant could have obtained a positive outcome but the counterfactual could have been higher, thus regret (negative outcome), etc.

Results, you need tables for the regression analyses (to be reported in supplementary material), and highlight in the text only relevant aspects of the regression without details about parameters etc. this is the standard way to report regression results.

Review form: Reviewer 2

Recommendation

Major revision is needed (please make suggestions in comments)

Scientific importance: Is the manuscript an original and important contribution to its field?

Good

General interest: Is the paper of sufficient general interest?

Good

Quality of the paper: Is the overall quality of the paper suitable?

Good

Is the length of the paper justified?

Yes

Should the paper be seen by a specialist statistical reviewer?

No

Do you have any concerns about statistical analyses in this paper? If so, please specify them explicitly in your report.

Yes

It is a condition of publication that authors make their supporting data, code and materials available - either as supplementary material or hosted in an external repository. Please rate, if applicable, the supporting data on the following criteria.

Is it accessible?

Yes

Is it clear?

Yes

Is it adequate?

Yes

Do you have any ethical concerns with this paper?

No

Comments to the Author

Please see attached review. (See Appendix A)

Decision letter (RSPB-2019-2251.R0)

25-Nov-2019

Dear Dr El Zein:

Your manuscript has now been peer reviewed and the reviews have been assessed by an Associate Editor. The reviewers, AE, and I all find your paradigm to be innovative and your results to be interesting. As you will see, however, the reviewers and the Editors have raised some concerns with your manuscript and we would like to invite you to revise your manuscript to address them. The reviewers' comments (not including confidential comments to the Editor) and the comments from the Associate Editor are included at the end of this email for your reference.

Research ethics:

Use of animals and field studies:

If your study uses animals please include details in the methods section of any approval and licences given to carry out the study and include full details of how animal welfare standards were ensured. Field studies should be conducted in accordance with local legislation; please

include details of the appropriate permission and licences that you obtained to carry out the field work.

If you wish to submit your data to Dryad (<http://datadryad.org/>) and have not already done so you can submit your data via this link [http://datadryad.org/submit?journalID=RSPB&manu=\(Document not available\)](http://datadryad.org/submit?journalID=RSPB&manu=(Document%20not%20available)), which will take you to your unique entry in the Dryad repository.

Please submit a copy of your revised paper within three weeks. If we do not hear from you within this time your manuscript will be rejected. If you are unable to meet this deadline please let us know as soon as possible, as we may be able to grant a short extension.

Best wishes,

Dr Sarah Brosnan

Associate Editor

Board Member: 1

Comments to Author:

This paper uses a novel experimental paradigm to test whether people make risky decisions preferentially in a collective setting. The results seem a little less strong than currently emphasised by the authors, so I would recommend more caution in interpretation. Both

reviewers make helpful suggestions and ask for clarification on a number of points, including whether the experimental design really accounts for collective behaviour.

Reviewer(s)' Comments to Author:

Referee: 1

Comments to the Author(s)

The authors report the results of an interesting experiment on the choice to either make choices in isolation or be part of a collective decision. Participants were asked to choose between two risky gambles. Experienced and anticipated regret played a role in the decision to be part of a collective choice.

This is an interesting and original experimental work.

Comments:

My main concern is about the fact that the behavior of others in the experiment was not a collective behavior, thus they didn't know that they were part of a group. This would have potentially affected their behavior. Thus, the participants are observing the sum of individual and not collective behavior.

"We note here that our aim is not to disentangle regret and loss from one another; instead we acknowledge that they are strongly related negative emotions, and focus on examining their hypothesized impact on the choice to join a collective." This sentence is not clear, Please rewrite it in a more comprehensive way

In experiment 1 participants experienced choices made by others in a one-player condition. This might have affected the data. A comparison of the results from Exp1 and Exp 2 is missing.

Experiment 2, not clear "when deciding to play in a group or alone and in when they decided to play in group what was their choice? Motivation for this is not clear. Do they choose before or after observing the choice of the others? Is their choice part of the collective choice?

How different is the behavior of the group compared with the one of each participant and how this correlates with the choice to join the collective. For each choice parameter, i.e. ev, regret and risk, you should estimate individual differences with the group.

Experiment 2, I would suggest to show the choice data of the reference group, i.e. estimate how many risky vs. safe choices they made independently from the individual choice (i.e. with already 3) and the same analysis for anticipated regret choices.

Do participants first see the lotteries and then decide if join or not the group? This would have been a nice extension of the study and would have linked this study with the literature about confidence.

Group choice. Not clear if COV and COM are lagged variables, i.e. $t-1$. Not clear how you computed the COM. So choice at time t as a function of lagged variables. It's confusing because you call the "most recent outcome" the current outcome, and then use it as a lagged variable. This section is very confusing. You can still disentangle the effect of the outcome and the effect of the counterfactual outcome on subsequent choice. E.g. the participant could have obtained a positive outcome but the counterfactual could have been higher, thus regret (negative outcome), etc.

Results, you need tables for the regression analyses (to be reported in supplementary material), and highlight in the text only relevant aspects of the regression without details about parameters etc. this is the standard way to report regression results.

Referee: 2

Comments to the Author(s)
Please see attached review.

Author's Response to Decision Letter for (RSPB-2019-2251.R0)

See Appendix B.

RSPB-2019-2251.R1 (Revision)

Review form: Reviewer 1

Recommendation

Accept as is

Scientific importance: Is the manuscript an original and important contribution to its field?

Excellent

General interest: Is the paper of sufficient general interest?

Excellent

Quality of the paper: Is the overall quality of the paper suitable?

Good

Is the length of the paper justified?

Yes

Should the paper be seen by a specialist statistical reviewer?

No

Do you have any concerns about statistical analyses in this paper? If so, please specify them explicitly in your report.

No

It is a condition of publication that authors make their supporting data, code and materials available - either as supplementary material or hosted in an external repository. Please rate, if applicable, the supporting data on the following criteria.

Is it accessible?

No

Is it clear?

N/A

Is it adequate?

N/A

Do you have any ethical concerns with this paper?

No

Comments to the Author

The authors assessed all my concerns.

Decision letter (RSPB-2019-2251.R1)

18-Feb-2020

Dear Dr El Zein

I am pleased to inform you that your manuscript entitled "Joining a group diverts regret and responsibility away from the individual" has been accepted for publication in Proceedings B.

Open Access

Paper charges

Sincerely,

Dr Sarah Brosnan
Editor, Proceedings B

Associate Editor:

Board Member: 1

Comments to Author:

I think this will paper make an interesting and novel contribution to the literature.

Board Member: 2

Comments to Author:

The revisions look extensive and in line with the concerns. However, the significant changes to the method need to now be seen by the reviewers to ensure that the original assessment still stands.

Appendix A

Review of RSPB-2019-2251

This article employs a risky choice paradigm to explore the possibility that negative consequences encourage us to move from making decisions alone to making decisions as part of a group, where responsibility for decisions can be distributed across group members. The authors suggest that deciding in a collective may provide a ‘protective shield’ against negative emotions, which may ultimately help enable individuals to be more objective (i.e., rely more on EV) about future choices. The predictions are well motivated, and the evidence is largely supportive of the hypotheses, though sometimes weak and occasionally inconclusive. There may also be questions about the approach to, or the detail of, analyses that might be worth revisiting to address. In addition, there may be some problems of interpretation, particularly with respect to issues surrounding the relationship between expected value difference, which is widely used as a criterion for choice quality, and the use of EV difference as a (weak) stand-in for the construct of anticipated loss.

Specific comments:

p.5, mid The use of expected value as a stand-in for anticipated loss is indirect at best, but also problematic. First, differences in EV are typically adopted as the primary variable for optimizing choice, and, second, the measure equally represents the contribution of gain and loss outcomes as weighted by their probabilities. Moreover, potential loss is not clearly specified by a given EV; the same EV may be associated with a wide variety of potential losses, especially as outcome probabilities change. Using EV also ambiguates whether the focus is on the likelihood of a loss or the potential size of a loss, both of which were readily available to be used in this type of trial-based analysis. Moreover, EV is also recognized later in the manuscript to describe the quality of the decision with less reliance attributed to ‘disrupt[ion in] loss anticipation’ (p.18,top). This usage at minimum denies the contributing roles of probability and gain outcomes to what is being assessed, which seems pivotal when explicitly making a case about the role of losses. Couldn’t this fundamental problem be alleviated by using one of the easily available more direct measures of anticipated loss?

p. 5 bottom The authors note that regret and loss (i.e., EV?) are known to be highly related, and that no attempt is made in the studies to disentangle them. This leads one to wonder later on what to make of findings wherein relationships are found for changes in EV but not in AR. Is there a better way to deal with this?

p. 7, top Choices were completed in four blocks of 12 trials each. Were lottery pairs unique across all 48 trials, or at least within blocks? Also, how was this structure incorporated, if at all, in the analyses? If not, how was within subjects vs between subjects variance separated in analyses (p. 9)?

p.7, bottom If space permits, it would be helpful to include the table showing all lotteries rather than referring readers to another publication.

p. 9, bottom Although admittedly a personal preference, it might be helpful to avoid these abbreviations (COV, COM, CC, CS) as it is difficult to remember them later in the article (esp. CC and CS—perhaps consider choosing more obvious initials?). Also, it would be helpful to provide an example for the transformation yielding the Current Outcome Magnitude (and a mention of why the values are different in the positive and negative domains).

p. 12, mid In general, it would seem preferable to deal more directly with the number of participants that never changed their preference to make decisions alone or in a group. As shown in Figure 2a, it appears that just over half of all participants (~65/125 and ~225/451) never switched their preference to play alone or in a group, so presumably, they were not persuaded by any of the manipulated variables. This should be pointed out more clearly so that the potential impact of previous outcomes can be put into perspective. Moreover, it may be worth considering removal of these participants in some analyses to specifically focus on the type of participant who can be persuaded to switch. Provided it is clear that this represents only a subset of people, the analysis might be clearer (and perhaps less likely to suffer from concerns about ceiling or floor effects). Either way, it might also be worth considering a metric based on stay versus switch behavior as it may better focus the analysis on what influences us to *change* behavior with respect to whether we want to be alone or in a group. Along these lines, it would also be useful to change the y-axis on Figures 2B and 2C in order to be able to compare the proportion of changes using a comparable reference point for all conditions. Finally, this change in focus to switching behavior would also allow for consideration (or at least mention) of well-known inertia effects wherein, all else equal, people are more likely to stay where they are. Individual difference information, such as proportion of participants who overall have more switches to group after a loss vs a gain, would also show how typical it is that a previous loss is a predictor of the tendency to join a group.

p. 12, bottom In Figure 2C, only the outcomes associated with losses are presented. This makes sense given the focus of this manuscript. However, it may hide a more general question which is: when do previous outcomes influence people to change their behavior to make decisions alone or in a group? What did the data look like for gain outcomes? Is it possible that people want to be in groups after experiencing extreme outcomes of either valence? Or perhaps, the better the outcome, the more we want to make our decisions alone. These data would be instructive for isolating concern about losses or diffusion of responsibility as primary or overarching drivers of the desire to decide in a group.

p. 13, top Characterizing findings here as a 'win–stay lose–change effect in both directions' doesn't seem to make sense because the pattern of 'lose-stay win-change' (line 310) is the opposite of that strategy. It is commendable to acknowledge that the focus on switch and stay has fundamental value in this context, but characterized relative to whether one has previously been alone or in a group.

p. 15, mid It would be useful to report how highly correlated the EV and AR measures were. Does it make sense to include them both?

p. 15. Bottom Consider removing reference to the (technically inaccurate usage) 'very significant finding' to focus instead on effect size.

p. 16, top and elsewhere The extensive listing of statistical findings might be more easily processed in a table as this makes it rather difficult to find the associated text.

p. 17, bottom The interpretation of the findings with respect to EV differences after experiencing a loss do not seem to match the description in text. How does the graph show that "Experiencing negative versus positive outcomes alone, but not in a group, reduced the EV parameter for the next lottery choice?" (Does the parameter represent the relation in EV use on the previous trial, and if not, in what sense might it reflect a reduction?) I may well be misinterpreting, but Figure 3d seems to show that the EV parameter (which could represent anticipated gain as easily as anticipated loss) is substantial but of similar size across the alone, group majority, and group minority conditions after receiving a previous

loss outcome but that it differs markedly across conditions when a participant previously experienced a gain outcome, being most predictive of choice when alone, intermediate when in the group majority, and small when in the minority. At least nominally, the EV parameter after a loss was slightly higher when alone than in a group. Please clarify if that interpretation is incorrect. If that is what was found, doesn't it suggest that the role of EV is most dramatically different across alone/group conditions after gain outcomes with only slight differences across conditions after loss outcomes? If so, how does this contribute to thinking about valenced outcomes and what makes us want to make decisions alone vs in a group?

p. 18, line 466 What is meant by 'their subsequent choice [being] unimpaired'? Is the impairment *less* reliance on EV (which seems to go with the anticipated loss interpretation) or *more* reliance on EV which is what standard economic theory would advise?

p.19, bottom It seems a rather tenuous connection between less role for EV and AR after a loss to 'feel[ing] less responsible.' Perhaps a bit more could be said about the logical reasoning or presumed equivalence. Are there other possibilities?

Appendix B

We would like to start by informing the editor and reviewers about an important issue that came to our attention while revising the paper. We realized that in our Experiment 2 (replication of Experiment 1 with a few changes), only half of the lottery structures (24 out of 48 total from Coricelli et al., 2005) were actually presented to each participant. This was due to an error in the randomization script of the experiment. We sincerely apologize for this error. The error does not, however, falsify the analyses done, as it only reduced the variety in the lotteries shown. Nevertheless, we think it is an incorrect experimental design and therefore we reconducted the online experiment 2 after fixing the issue, this time with all 48 rounds. We now report all the analyses for Experiment 1 and (the correct) Experiment 2. The results of the initial Experiment 2 (now referred to as Experiment 2b) are reported in supplementary materials for completeness. The main findings and conclusions from Experiment 2 remain unchanged (hence, the abstract is unchanged).

The data of all three experiments is now shared on Dryad, with the explanation of each of the variables and what happened with Experiment 2b. The following link can be used in the review process:

<https://datadryad.org/stash/share/aX4bWZi0OwqobQvWTmLnIXNwEU0McUMw1BfX4Og0Li4> (the actual link that is cited in the manuscript is not available during the review process <https://doi.org/10.5061/dryad.gqnk98shr>)

Please find below our point-by-point responses to the editor and the two reviewers' comments.

Associate Editor

- This paper uses a novel experimental paradigm to test whether people make risky decisions preferentially in a collective setting. The results seem a little less strong than currently emphasised by the authors, so I would recommend more caution in interpretation. Both reviewers make helpful suggestions and ask for clarification on a number of points, including whether the experimental design really accounts for collective behaviour.

We hope to have addressed all the reviewers' comments to clarify our results and report more cautious interpretations as recommended by the editor. We also hope to have clarified and corrected as suggested that our experimental design tackled what makes an individual join a group, but did not account for an emerging collective behaviour. We describe the way we did that in details in the answers to the reviewers below.

Referee 1

- The authors report the results of an interesting experiment on the choice to either make choices in isolation or be part of a collective decision. Participants were asked to choose between two risky gambles. Experienced and anticipated regret played a role in the decision to be part of a collective choice.

This is an interesting and original experimental work.

We thank the reviewer for acknowledging the originality of the paper and for providing useful suggestions that improved the paper.

- My main concern is about the fact that the behavior of others in the experiment was not a collective behavior, thus they didn't know that they were part of a group. This would have potentially affected their behavior. Thus, the participants are observing the sum of individual and not collective behavior.

We thank the reviewer for pointing out to this issue. We agree that the behaviour observed here is a sum of individual behaviours, and that the vocabulary we used ('collective decisions') incorrectly implied collective behaviour. Our design allowed us to maintain control over 'other's behaviour' as an independent variable, however not accounting for an emerging collective behaviour of autonomous interacting individuals. We have now changed the vocabulary we use throughout the manuscript and clarified this point:

Title: 'Joining a group diverts regret and responsibility away from the individual'. (L.1-2, P1)

Introduction: 'Using a task in which decisions were costly and could elicit highly negative emotions via factual (loss) and counterfactual (regret) outcomes (Coricelli et al., 2005), we asked participants to choose, on a trial-by-trial basis, whether to decide alone or contribute their opinion to a majority vote-count in a group of five. Such majority votes offer an empirically controlled way of sharing responsibility with other individuals for a choice's outcome.' (L.78-83, P.4).

Methods: 'This design allowed us to maintain control over "other's behaviour" as an independent variable. It is important to underscore here that, the aggregation of the votes of such virtual group does not constitute an emergent collective. The design we chose here is appropriate to test our hypothesis about how the context of deciding alone vs with others affects individual choices but and importantly, our hypothesis does not involve examining the emergent collective behaviour of a group of interacting individuals.' (L.145-150, P.6).

Discussion: 'We note that our design was not meant to and did not permit accounting for the collective behaviour of interacting individuals. Our objective here was to take the individual's perspective to address what motivates people to join a group. Therefore, we chose a majority vote procedure using previously gathered responses from participants in a separate experiment thereby maintaining control over other 'group members' behaviour and allowing to address majority and minority impact on the choice to play alone or join a group. Future research is necessary to examine whether our results replicate under the less controlled design of an entire group of autonomous interacting individuals.' (L423-431, P14-15).

- "We note here that our aim is not to disentangle regret and loss from one another; instead we acknowledge that they are strongly related negative emotions, and focus on examining their hypothesized impact on the choice to join a collective." This sentence is not clear, Please rewrite it in a more comprehensive way

We removed the unclear sentence and added:

'Importantly, our design offered a powerful quantitative method to test how the valence and magnitude of experienced outcomes, combining loss and regret, influenced participants' choices to decide alone or in a group.' (L86-89, P4).

The point was to explain that we will use a combination of experienced loss and regret to quantify experienced outcomes rather than separate them, for the reasons explained below in response to another point by the reviewer (P. 6 of the rebuttal).

- In experiment 1 participants experienced choices made by others in a one-player condition. This might have affected the data. A comparison of the results from Exp1 and Exp 2 is missing.

We understand the reviewer's concern about previous responses used in Experiment 1 played in the alone condition only. As explained in the previous comment, we now have clarified our vocabulary to say that we are investigating individual behaviours in a collective context, rather than an interactive collective behaviour. From this point of view, Experiments 1 and 2 are exactly similar from the perspective of the participant. As shown in the supplementary tables, the majority vote choice is similar in both experiments, but more variable in Experiment 2. Here, as the reviewer suggests, we have addressed the problem by directly comparing the results of experiment 1 and 2 using a meta-analytic approach. This meta-analytical approach allowed to compare and combine the 2 studies. Moreover, throughout our reports of the results, we report any possible difference between Exp 1 and Exp 2.

- Experiment 2, not clear "when deciding to play in a group or alone and in when they decided to play in group what was their choice? Motivation for this is not clear. Do they choose before or after observing the choice of the others? Is their choice part of the collective choice?"

We apologise for this misunderstanding, and we hope we have now made this as salient as we can to avoid misunderstanding. In both experiments, participants chose if they want to play alone or in group at the beginning of the trial, before they see the lotteries and the choices of others (Fig 1a, timeline for one trial shown). In both experiments, from the perspective of the participant, their choice is part of the collective choice. In experiment 1, the collective choice was based on the collective choice of 11 fixed participants (deception was involved). In experiment 2, their choice was really part of the collective choice (and the choices from 4 randomly drawn previous participants). This information is detailed in the methods section (L132-150, P.6). We also added a sentence in the results section to provide the motivation for why the design was done this way (i.e., group choice followed by lottery choice):

'In each round, before seeing the lotteries, participants first chose whether they wanted to play an upcoming lottery choice alone or in a group. On every trial, this decision to join the group or not was made before the lotteries were seen. This ensured that we could assess the direct influence of previous experienced outcome on this choice independently from the upcoming lottery structure.' (L.269-273, P.10)

- How different is the behavior of the group compared with the one of each participant and how this correlates with the choice to join the collective. For each choice parameter, i.e. ev, regret and risk, you should estimate individual differences with the group.

We have added a supplementary table with all the 48 rounds details (Supplementary table 1). We provide for each lottery pair, and thereby for each value of expected value and anticipated regret (reported in the table), the percentage of time a lottery was chosen over the other for both the group decisions and the individual decisions.

1- Group choice EXP1	2-Exp 2 % Group Choice Lottery1	3-Exp 2 % Group Choice Lottery2	4-Exp 2 % Individual Choice Lottery 1	5-Exp 2 %Individual Choice Lottery 2	6-Exp 2 mean % Lottery 1 reference group
---------------------------------	---------------------------------	---------------------------------------	--------------------------------------	--

- 1- The first column reports for each lottery pair the gamble chosen by the fixed pilot group.
- 2- The second column reports the percentage of times where Lottery 1 was chosen by the randomly drawn group.
- 3- Same as the third column for Lottery 2.
- 4- The fourth column reports the percentage of **individual** choices of Lottery 1.
- 5- Same as the fourth column for Lottery 2.
- 6- The 6th column reports the % choice of the Lottery 1 for the Reference group, as asked for in the next comment.

This gives an estimate of how the group responses were represented and how individual responses were given. The difference between a participant's behaviour and the group's behaviour is made visible to the participant on a trial by trial basis, when they choose to play in group: they see the chosen gamble at the end of the trial, which either corresponds to their individual choice or not. This behaviour does indeed correlate with the choice to join a group, as shown in our analysis with the group status (L.341-343, P.12): when in the group majority (similarity in behaviour between participant and group – agreement with the group), participants tended to join the collective more than when they were in the group minority (difference between the individual behaviour and the group – disagreement with the group).

- Experiment 2, I would suggest to show the choice data of the reference group, i.e. estimate how many risky vs. safe choices they made independently from the individual choice (i.e. with already 3) and the same analysis for anticipated regret choices.

Participants did not see all the group members' choices, but only the final group choice in the group condition that includes their own choice in Experiment 2. Therefore, we do not include the choice data of the reference group as a regressor in our analyses. What counts for the participant is whether he/she agreed (group majority) or disagreed (group minority) with the group.

For each participant and each trial, a random selection of previous subjects constituted the group with whom the participant engaged. Therefore, the reference group data was different for each participant. To show the choice data of the reference group as asked for by the

reviewer, we therefore make a mean of percentage of choices of Lottery 1 over Lottery 2 among the 4 previous participants. This is shown in the table described in a previous answer to the reviewer (column 6, P.4 of the rebuttal) with the corresponding Expected Value and Anticipated regret differences values.

- Do participants first see the lotteries and then decide if join or not the group? This would have been a nice extension of the study and would have linked this study with the literature about confidence.

In Figure 1a, the timeline for each trial for both experiments is shown: participants first choose if they want to play alone or as a group, then see the lotteries for the round. Participants therefore did not first see the lottery and then decide whether to join or not group. We have now clarified our motivation for this in the results section, as described above (P.3 of the rebuttal). We agree with the reviewer this would be a very nice extension of our current study and hope this will be addressed in future experiments. Here, as the first experimental paradigm with a trial-by-trial choice to play alone or as a group, we wanted to focus on the influence of experienced outcome on this choice, and how lottery decisions are influenced by whether one decided to play alone or as a group.

- Group choice. Not clear if COV and COM are lagged variables, i.e. $t-1$. Not clear how you computed the COM. So choice at time t as a function of lagged variables. It's confusing because you call the "most recent outcome" the current outcome, and then use it as a lagged variable.
This section is very confusing. You can still disentangle the effect of the outcome and the effect of the counterfactual outcome on subsequent choice. E.g. the participant could have obtained a positive outcome but the counterfactual could have been higher, thus regret (negative outcome), etc.

We apologize for not being clear about the explanation of the regressors used and how we computed the magnitude.

We now explain in more details how the magnitude was computed in the methods sections L188-202. P.7-8.:

'For the Magnitude predictor, we transformed experienced outcome values into magnitude values only reflecting the change in magnitude between the outcomes. Outcome values (-400, -250, -150, -100, 100, 150, 250, 400) were transformed for negative values into (0, 150, 250, 300) and for positive values into (0, 50, 150, 300).

Negative values

- $0+(150 \text{ difference between } -400 \text{ and } -250)$
- $150+(100=\text{difference between } -250 \text{ and } -150)$
- $250+(50=\text{difference between } -150 \text{ and } -100)$

Positive values

- $0+(50 \text{ difference between } 100 \text{ and } 150)$

- $50+(100=\text{difference between } 150 \text{ and } 250)$
- $150+(150=\text{difference between } 250 \text{ and } 400)$

Due to the lottery structures, the magnitude of possible outcomes was slightly different for positive and negative outcomes.'

We realized thanks to the reviewers' comments that the description of our predictors was very confusing and therefore we changed all the vocabulary and now add whether the predictors are at t or t-1 (also in the supplementary tables reporting all the regression results). Indeed, valence and magnitude are lagged variables – at t-1 (see new description of variables in methods P. 7-9).

We did not have precise differential predictions for factual and counterfactual outcomes influences on the decision to play alone or in a group. We did this experiment to test how experienced outcomes - in a context where regret is possible – influence whether people choose to play alone or in a group. Therefore, we combine these two correlated measures in the current experimental design, to focus on the influence of experienced outcomes on joining a group. In the current design, there are not enough conditions in which as described by the reviewer, both outcomes and counterfactuals share the same valence but the counterfactual is higher. Future experiments explicitly designed to disentangle counterfactual from factual outcomes may allow to test whether they differentially impact the propensity to play alone or in a group.

- Results, you need tables for the regression analyses (to be reported in supplementary material), and highlight in the text only relevant aspects of the regression without details about parameters etc. this is the standard way to report regression results.

We thank the reviewer for suggesting this, which we agree made the results much easier to read. Supplementary tables (2 to 6) with all regression details are now added, while only essential statistical details are left in the text.

Reviewer 2

This article employs a risky choice paradigm to explore the possibility that negative consequences encourage us to move from making decisions alone to making decisions as part of a group, where responsibility for decisions can be distributed across group members. The authors suggest that deciding in a collective may provide a 'protective shield' against negative emotions, which may ultimately help enable individuals to be more objective (i.e., rely more on EV) about future choices. The predictions are well motivated, and the evidence is largely supportive of the hypotheses, though sometimes weak and occasionally inconclusive. There may also be questions about the approach to, or the detail of, analyses that might be worth revisiting to address. In addition, there may be some problems of interpretation, particularly with respect to issues surrounding the relationship between expected value difference, which is widely used as a criterion for choice quality, and the use of EV difference as a (weak) stand-in for the construct of anticipated loss.

We thank the reviewer for providing useful comments and suggestions that improved the paper. We followed the reviewer's advice to not use the EV as stand-in for anticipated loss, and thank the reviewer for pointing that the relation between EV and anticipated loss is not as straightforward as we were suggesting.

Specific comments:

- p.5, mid The use of expected value as a stand-in for anticipated loss is indirect at best, but also problematic. First, differences in EV are typically adopted as the primary variable for optimizing choice, and, second, the measure equally represents the contribution of gain and loss outcomes as weighted by their probabilities. Moreover, potential loss is not clearly specified by a given EV; the same EV may be associated with a wide variety of potential losses, especially as outcome probabilities change. Using EV also ambiguates whether the focus is on the likelihood of a loss or the potential size of a loss, both of which were readily available to be used in this type of trial-based analysis. Moreover, EV is also recognized later in the manuscript to describe the quality of the decision with less reliance attributed to 'disrupt[ion in] loss anticipation' (p.18,top). This usage at minimum denies the contributing roles of probability and gain outcomes to what is being assessed, which seems pivotal when explicitly making a case about the role of losses. Couldn't this fundamental problem be alleviated by using one of the easily available more direct measures of anticipated loss?

We thank the reviewer for pointing out that the relation between EV and anticipated loss is not as straightforward as we were claiming. We have stopped using 'loss anticipation' and now rather use 'maximizing the expected value' and 'optimizing choice', which is, we agree with the reviewer, the clearer interpretation of EV.

We stopped using 'anticipated loss' throughout the text and changed in the introduction:

'Moreover, individuals choose not only options that optimize their choice by maximizing the expected value, but also those that minimize future regret' (L.58, P.3).

'Using computational behavioral analysis, we assessed whether the previously established influence of expected value and anticipated regret on individual lottery decisions changes when the participant acted as a member of a group.' (L.83-86, P.4).

- p. 5 bottom The authors note that regret and loss (i.e., EV?) are known to be highly related, and that no attempt is made in the studies to disentangle them. This leads one to wonder later on what to make of findings wherein relationships are found for changes in EV but not in AR. Is there a better way to deal with this?

We apologise for the confusion. In p.5, we were referring to *experienced* loss and regret – as we compute one common measure to experienced outcomes. We have now clarified this distinction in the introduction. Indeed, the analyses on the previous outcomes influence on lottery choice seem to have influenced choice optimization (with the corrected use of expected value) rather than anticipated regret.

- p. 7, top Choices were completed in four blocks of 12 trials each. Were lottery pairs unique across all 48 trials, or at least within blocks? Also, how was this structure incorporated, if at all, in the analyses? If not, how was within subjects vs between subjects variance separated in analyses (p. 9)?

As requested by the reviewer, all details for the 48 lottery pairs are now reported in supplementary table 1. The structure of the lotteries determines EV and AR predictors, as shown in the table next to each lottery pair. Among the 48 lottery pairs, there were 43 unique pairs of lottery. The 48 combinations shown in the new supplementary table 1 were all shown in a random order for every participant, and randomly separated in 4 blocks – In the methods section, we write: ‘Pairs of lotteries were shown in a randomized order for each participant.’ (L. 164-164, P7).

The structure is incorporated on a trial by trial basis with the associated expected value and anticipated regret value for each lottery. These values that derive from the lottery structure, are used as predictors of the lottery choice. Subjects were entered as random intercepts in the mixed models as described in the methods section.

p.7, bottom If space permits, it would be helpful to include the table showing all lotteries rather than referring readers to another publication.

We thank the reviewer for this suggestion. Lottery pairs are now reported in supplementary table 1, with all the details for each lottery pair.

- p. 9, bottom Although admittedly a personal preference, it might be helpful to avoid these abbreviations (COV, COM, CC, CS) as it is difficult to remember them later in the article (esp. CC and CS—perhaps consider choosing more obvious initials?). Also, it would be helpful to provide an example for the transformation yielding the Current Outcome Magnitude (and a mention of why the values are different in the positive and negative domains).

We apologize for using unclear abbreviations and variables names, as well as being unclear about how we computed the magnitude variable, and thank the reviewer for noting this. We followed the reviewer’s suggestion and did not use abbreviations in our revised version, but always use: Valence, Magnitude, Current or Previous Condition, and Status. We now explain in more details the magnitude transformation (L.188-202, P7-8):

‘For the Magnitude predictor, we transformed experienced outcome values into magnitude values only reflecting the change in magnitude between the outcomes. Outcome values (-400, -250, -150, -100, 100, 150, 250, 400) were transformed for negative values into (0, 150, 250, 300) and for positive values into (0, 50, 150, 300).

Negative values

- $0 + (150 \text{ difference between } -400 \text{ and } -250)$
- $150 + (100 = \text{difference between } -250 \text{ and } -150)$

- $250+(50=\text{difference between } -150 \text{ and } -100)$

Positive values

- $0+(50 \text{ difference between } 100 \text{ and } 150)$
- $50+(100=\text{difference between } 150 \text{ and } 250)$
- $150+(150=\text{difference between } 250 \text{ and } 400)$

Due to the lottery structures, the magnitude of possible outcomes was slightly different for positive and negative outcomes.'

- p. 12, mid In general, it would seem preferable to deal more directly with the number of participants that never changed their preference to make decisions alone or in a group. As shown in Figure 2a, it appears that just over half of all participants (~65/125 and ~225/451) never switched their preference to play alone or in a group, so presumably, they were not persuaded by any of the manipulated variables. This should be pointed out more clearly so that the potential impact of previous outcomes can be put into perspective. Moreover, it may be worth considering removal of these participants in some analyses to specifically focus on the type of participant who can be persuaded to switch. Provided it is clear that this represents only a subset of people, the analysis might be clearer (and perhaps less likely to suffer from concerns about ceiling or floor effects).

We added a sentence in the text to highlight this: 'It is important to note that about half of participants consistently chose to play alone or in group (more than 90% of the times - Exp 1 60/125, Exp 2 269/496)– and therefore were not affected by experienced outcomes.' (L274-277, P.10)

And also discuss this in discussion:

'These results suggest that 1) there are baseline, trait-like preferences to play alone or in group that are independent from experienced outcomes' (L449-451, P.15).

As suggested by the reviewer, analyses on the group choice after removing these participants were done, we report in supplementary table 5 the statistics for meta-analytical results, that replicate the findings on the whole sample .

We show here the descriptive plots when doing the analyses after the exclusion:

We prefer to keep the whole sample for our main analyses rather than half of the sample: it is relevant to include those who consistently played alone or in group for the analyses on the lottery choice, but also for the Group status analyses (for those who played in a group consistently).

- Either way, it might also be worth considering a metric based on stay versus switch behavior as it may better focus the analysis on what influences us to change behavior with respect to whether we want to be alone or in a group. Along these lines, it would also be useful to change the y-axis on Figures 2B and 2C in order to be able to compare the proportion of changes using a comparable reference point for all conditions. Finally, this change in focus to switching behavior would also allow for consideration (or at least mention) of well-known inertia effects wherein, all else equal, people are more likely to stay where they are.

As suggested by the reviewer, for clarity of presentation, we now represent the data on Figure 2 as stay/switch behaviour. We also explain the data in those words in the manuscript. We mention (results section) and discuss (discussion section) as suggested – the status quo/inertia effect.

Results: 'A main effect of previous condition (see full statistical details in supplementary tables 2, 3, and 4, all $z > 8.96$, all $p < 0.001$), reflected the fact that some participants chose to play predominantly alone or in a group (as shown in Fig. 2a), but also a sort of inertia effect whereby people stick to their previous decision. Indeed, even when excluding participants who invariably chose to stay alone or in group – the main effect of previous condition remained significant (see supplementary table 5, meta-analytically $z = -3.9$, $p < .001$).' (L.295-301, P.10-11).

Discussion: 'In parallel to the influence of valence, participants tended to repeat their choice to stay alone, revealing an inertia/status-quo effect (Samuelson & Zeckhauser, 1988). This was also the case when people were playing in group. Moreover, the tendency to repeat one's choice was observed in participants who regularly switched their choice from playing alone to playing in group. These results suggest that 1) there are baseline, trait-like preferences to play alone or in group that are independent from experienced outcomes; 2)

even when people vary their choice, a tendency to repeat the previous choice may show that participant exploit the same option before switching. Interestingly, the emotion of regret may induce a status-quo bias (A. Nicolle, Fleming, Bach, Driver, & Dolan, 2011) – suggesting that status-quo in the decision to play alone or in group may be a strategy to reduce regret in the context of the current study.’ (L.445-456, P.15).

- Individual difference information, such as proportion of participants who overall have more switches to group after a loss vs a gain, would also show how typical it is that a previous loss is a predictor of the tendency to join a group.

Below is individual difference information in Experiments 1 and 2 on the tendency to switch to group (from alone) after negative vs positive outcomes. The difference in the proportion of joining groups between negative and positive experienced outcomes is shown on the x-axis. The histograms show the number of participants who show a positive difference (black square), i.e., switched more frequently from playing alone to in group after negative vs positive outcomes. A student t-test on these proportions confirms the mixed models analyses showing that participants significantly joined groups more often after experiencing negative vs positive outcomes alone (EXP 1 $T=2.21$, $p=0.02$, EXP 2 $T=2.89$, $p=0.004$). We have added this as a supplementary figure (supplementary figure 1), referred to in the manuscript:

‘Participants were more likely to switch their choice after a negative vs a positive outcome experienced alone (all $z>3.11$, all $p<0.04$, also see individual difference information on the propensity to switch from alone to group based on valence in supplementary figure 1), or in a group (all $z>3.57$, all $p<.002$) (Fig 2b).’ (L.307-310, P.11).

- p. 12, bottom In Figure 2C, only the outcomes associated with losses are presented. This makes sense given the focus of this manuscript. However, it may hide a more general question which is: when do previous outcomes influence people to change their behavior to make decisions alone or in a group? What did the data look like for gain outcomes? Is it possible that people want to be in groups after experiencing extreme outcomes of either valence? Or perhaps, the better the outcome, the more we want to make our decisions alone. These data would be instructive for isolating concern about losses or diffusion of responsibility as primary or overarching drivers of the desire to decide in a group.

We have followed the reviewer's request, and now included the data for gain outcomes in Figure 2C. Even though the slope looks steeper in the negative domain, the effect is in the same direction in the gain domain: the better the outcome, the more participants stay alone. There is no significant evidence that the propensity to play in group based on magnitude is only in the negative domain. Given our results, we can only conclude that the more negative the outcome (even in the gain domain) the more people join groups. We have adapted the text accordingly:

'The choice to join the group was sensitive to the magnitude of outcomes experienced alone (meta-analytically $z=2.36$, $p=0.01$ – Exp1 $p=0.17$, Exp 2 $p=0.04$). In other words, the less positive/more negative the outcome, the more likely were the participants to join a group on the next round.' (L.317-321, P.11).

- p. 13, top Characterizing findings here as a 'win–stay lose–change effect in both directions' doesn't seem to make sense because the pattern of 'lose-stay win-change' (line 310) is the opposite of that strategy. It is commendable to acknowledge that the focus on switch and stay has fundamental value in this context, but characterized relative to whether one has previously been alone or in a group.

We have now changed as suggested the graphs to 'switch' behaviour in order to clarify what we report.

On line 310 of the first submitted manuscript: we meant that, when in the group, it is less likely to join a group after a negative outcome, i.e more likely to switch to playing alone. Therefore this also a lose-switch to alone, win stay in group effect. Explaining this as a switch behaviour as suggested by the reviewer should clarify that both after experiencing outcomes alone or in group, participants tended to switch more after losses compared to wins.

- p. 15, mid It would be useful to report how highly correlated the EV and AR measures were. Does it make sense to include them both?

EV and AR are indeed correlated regressor ($r=0.7$) that have been used in the same regression to predict lottery choice previously (Coricelli et al., 2005; Mellers et al., 1999). It is important to include both of them as there is no clear way to combine them: EV, which is the established predictor of value-based choices, depends on probabilities and outcome; while AR ignores probabilities and focuses on an overall measure dependent on the comparison between anticipated factual and counterfactual outcome. They can accurately both be included in the model given the following:

- 1) Their variance inflation factor (VIF), that is calculated to deal with multicollinearity, is equal to 2.33. It is usually considered highly problematic to include correlated regressors if the values are higher than 10, and a threshold of 2.5 has also been suggested. Therefore, with $VIF=2.33$ we can include them both in our mixed models. When the two regressors are included to predict lottery choice, the output of the mixed model than should reflect the *unique* effect of each parameter.
- 2) In fact, the `glme` function in R uses an orthogonalization procedure to decorrelate the effects of different regressors, that we also manually performed to confirm that the

observed effects are unique contributions of the EV and AR regressors. We did a Gram Schmidt orthogonalization of the two regressors using the QR() and QR.Q functions in R. We then assessed the influence of orthogonalized EV and AR regressors, and obtained exactly the same statistical output from the initial model (see supplementary table 7, in comparison with supplementary tables 2 and 3).

We added this in our methods section:

' Δ EV and Δ AR are correlated choice predictors ($r=0.75$, $p<.001$), however 1) their variance inflation factor (VIF) is equal to 2.33 suggesting that it is acceptable to put them in the same general linear model. In the models, only the variance not accounted by the other predictor is reflected in the results. 2) the parametric regressors were QR Gram Schmidt orthogonalized (using QR() and QR.Q functions in R) to enter in our GLM the residuals of the regressors after removing the common variance to both predictors. This yielded to exactly similar results (see supplementary table 7) confirming that it is correct to include them both as predictors of the lottery choice, and showing that this orthogonalization procedure was performed automatically via the glmer R function.' (L.242-251, P.9)

- p. 15. Bottom Consider removing reference to the (technically inaccurate usage) 'very significant finding' to focus instead on effect size.

We absolutely agree on this comment and have removed this inaccurate usage.

- p. 16, top and elsewhere The extensive listing of statistical findings might be more easily processed in a table as this makes it rather difficult to find the associated text.

We agree and thank the reviewer for suggesting this. We now report all the statistical details in supplementary tables (Supplementary tables 1 to 6).

- p. 17, bottom The interpretation of the findings with respect to EV differences after experiencing a loss do not seem to match the description in text. How does the graph show that "Experiencing negative versus positive outcomes alone, but not in a group, reduced the EV parameter for the next lottery choice?"

We apologize for this, this sentence interpreted the results before taking into account the group status (majority/minority). An interaction between Valence at t-1 and EV appeared only when outcomes were experienced alone at t-1, not in group. We agree that referring to the figure 3d after this is confusing as the figure shows the result with the group status. We accordingly changed the text to clarify:

'Negative vs positive outcomes reduced the Δ EV parameter for the next lottery choice, only when experienced alone (Fig 3d), but critically not in group (see next paragraph on how this is different based on group status).' (L.388-391, P.13).

- (Does the parameter represent the relation in EV use on the previous trial, and if not, in what sense might it reflect a reduction?) I may well be misinterpreting, but Figure 3d seems to show that the EV parameter (which could represent anticipated gain as easily as anticipated loss) is substantial but of similar size across the alone, group majority, and group minority conditions after receiving a previous loss outcome but that it differs markedly across conditions when a participant previously experienced a gain outcome, being most predictive of choice when alone, intermediate when in the

group majority, and small when in the minority. At least nominally, the EV parameter after a loss was slightly higher when alone than in a group. Please clarify if that interpretation is incorrect. If that is what was found, doesn't it suggest that the role of EV is most dramatically different across alone/group conditions after gain outcomes with only slight differences across conditions after loss outcomes? If so, how does this contribute to thinking about valenced outcomes and what makes us want to make decisions alone vs in a group?

The triple interaction Previous condition(t-1) X Valence(t-1) X EV(t) can be interpreted in 2 complementary ways:

- 1) The way we interpreted the result as this was the hypothesis we were testing: The interaction between Valence at t-1 and EV present after playing alone not group (and after group minority), showing a decreased EV after negative vs positive outcomes.
- 2) The reviewer's interpretation: The interaction between Previous condition and EV present after positive but not negative outcomes. This suggests that after negative outcomes, whatever the condition (Alone, in group majority/minority) the EV is similar on the next trial. But after positive outcomes, participants rely less on EV if in group, and even less so if in the group minority. Running a model with the interaction between previous ConditionXEV separately after positive vs negative outcome indeed confirms this interpretation (interaction Previous ConditionXEV and Group StatusXEV only present after positive outcomes)

This suggests that choice optimization is decreased after positive outcomes experienced in a group as compared to alone, and experienced in a group majority as compared to group minority. Nevertheless, this result does not cancel or change the fact that after group majority, the EV parameter is stable whether after a positive or negative outcome – while the EV changes between positive and negative outcomes experienced alone or in a group minority. We pay attention to say that negative outcomes influenced subsequent choices more importantly when compared to *positive outcomes*.

We have now added this additional interpretation in supplementary results (P.13 in supplementary material).

- p. 18, line 466 What is meant by 'their subsequent choice [being] unimpaired'? Is the impairment less reliance on EV (which seems to go with the anticipated loss interpretation) or more reliance on EV which is what standard economic theory would advise?

We mean reliance on EV as the standard economy theory would advise. We followed the reviewer's advice to interpret reliance on EV as 'maximization the expected value' and 'optimizing choice', rather than 'anticipating loss' throughout the manuscript.

- p.19, bottom It seems a rather tenuous connection between less role for EV and AR after a loss to 'feel[ing] less responsible.' Perhaps a bit more could be said about the logical reasoning or presumed equivalence. Are there other possibilities?

Based on this comment we have changed this paragraph as follows, reducing the responsibility claim and suggesting an alternative explanation:

'From the shared responsibility hypothesis perspective (El Zein et al.,2019), this might suggest that participants feel less responsible for their choice when playing in a group. (...) Additionally, given that expected value influence also decreased in group, participants may have chosen to play in the group to exert less effort – a phenomenon known as social loafing (Simms & Nichols, 2014).' (L462-473, P.16).